# The formation and ventilation of an Oxygen Minimum Zone in a simple model for latitudinally alternating zonal jets

Eike E. Köhn[1,2], Richard J. Greatbatch[1,3], Peter Brandt[1,3], and Martin Claus[1]

[1]GEOMAR Helmholtz Centre for Ocean Research Kiel, Kiel, Germany
[2]Environmental Physics, Institute of Biogeochemistry and Pollutant Dynamics, ETH Zürich, Zürich, Switzerland
[3]Faculty of Mathematics and Natural Sciences, Christian Albrechts Universität zu Kiel, Kiel, Germany

**Correspondence:** Richard Greatbatch (rgreatbatch@geomar.de)

**Abstract.**

An advection-diffusion model coupled to a simple dynamical ocean model is used to illustrate the formation and ventilation of an oxygen minimum zone. The advection-diffusion model carries a passive tracer with a source at the western boundary and a Newtonian damping term to mimic oxygen consumption. The dynamical model is a non-linear $1\frac{1}{2}$-layer reduced-gravity model. The latter is forced by an annually oscillating mass flux confined to the near-equatorial band that, in turn, leads to the generation of mesoscale eddies and latitudinally alternating zonal jets at higher latitudes. The model uses North Atlantic geometry and develops a tracer minimum zone remarkably similar in location to the observed oxygen minimum zone in the Eastern Tropical North Atlantic (ETNA). This is despite the absence of wind forcing for a subtropical gyre and the shadow zone predicted by the ventilated thermocline theory. Although the model is forced only at the annual period, the model nevertheless exhibits decadal and multidecadal variability in its spun-up state. The associated trends are comparable to observed trends in oxygen within the ETNA oxygen minimum zone. Notable exceptions are the multi-decadal decrease in oxygen in the lower oxygen minimum zone, and the sharp decrease in oxygen in the upper oxygen minimum zone between 2006 and 2013.

## 1 Introduction

As the ocean warms due to anthropogenic forcing, oxygen levels in the ocean are expected to decrease, putting pressure on the ocean ecosystem and fisheries (Stramma et al., 2008; Keeling et al., 2010; Schmidtko et al., 2017; Breitburg et al., 2018; Oschlies et al., 2018). In particular, this deoxygenation process is expected to lead to an expansion of the oxygen minimum zones (OMZs). These are large, naturally occurring volumes of low-oxygen waters situated in the subsurface of the subtropical oceans (e.g., Karstensen et al. (2008)). Yet, the fate of the OMZs depends on the delicate interplay between biological oxygen consumption and physical oxygen supply. One critical aspect is therefore to understand how oxygen is supplied to OMZs and how this ventilation mechanism varies in time. Figure 1a shows the mean oxygen distribution (black contours) at roughly 500m depth in the tropical Atlantic Ocean. Prominent are the regions of low oxygen concentration in the eastern basin separated by a band of higher concentration centred on the equator, and the much higher levels of concentration found near the western boundary. The off-equatorial regions of low concentration at the eastern boundary are commonly associated with the shadow zones predicted by the theory for the ventilated thermocline (Luyten et al., 1983; Brandt et al., 2015) and are regions with

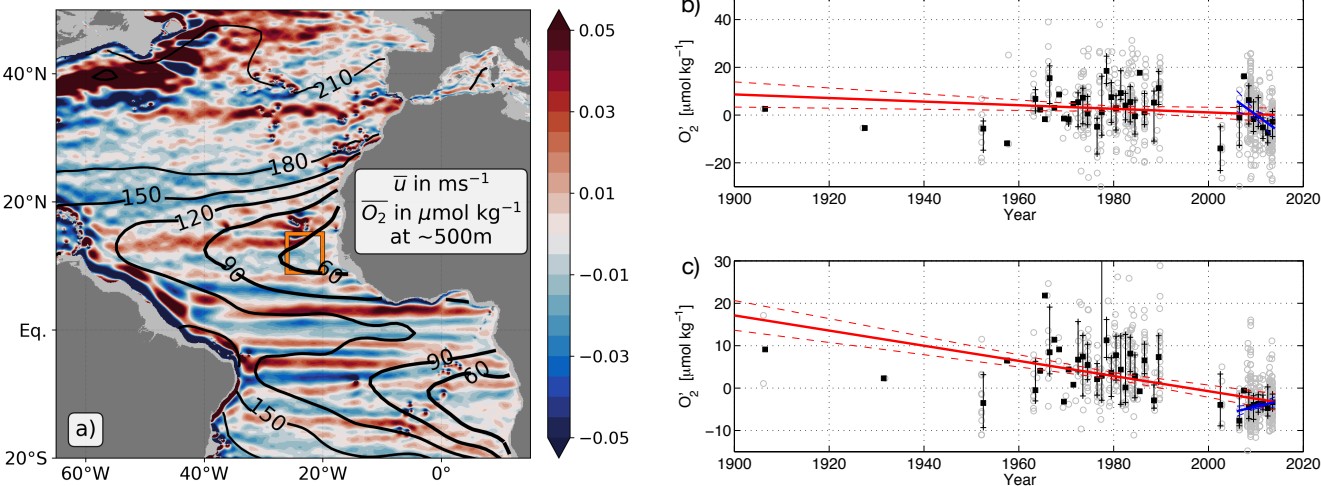

**Figure 1.** (a) Contours show the mean oxygen concentration ($\mu$mol kg$^{-1}$) in the tropical Atlantic at 500m depth (close to the deep oxygen minimum) as obtained from the World Ocean Atlas 2018 (Garcia et al., 2019). Colour shading shows the 1993-2016 time mean zonal velocity from the GLORYS12v1 reanalysis product averaged between the 454m and 541m depth levels (CMEMS, 2023). (b, c) Oxygen anomalies for the region $9°$–$15°$N, $20°$–$26°$W (orange box in panel a) for 150–300m (upper OMZ, panel b) and 350–700m (lower OMZ, panel c). Grey circles represent all available data, whiskers show the interquartile range of data within each year and the black squares annual medians. Trends are calculated using annual medians weighted by the square root of available data within each year for the period 1900–2013 (solid red line) and 2006–2013 (solid blue line). The dashed lines mark the standard errors of the trends. Panels b,c are taken from Fig. 19 in Brandt et al. (2015).

sluggish ocean circulation that are weakly ventilated. Brandt et al. (2015) discuss the Eastern Tropical North Atlantic (ETNA) OMZ, north of the equator, in some detail. These authors note the important role played by latitudinally alternating zonal jets for supplying oxygen to this region. These jets can be seen in Fig. 1a and appear as the alternating bands of eastward (red) and westward (blue) zonal velocity in a multi-year average. The importance of these jets has also been demonstated by Calil (2023) who found a much better representation of the Atlantic OMZs in a high resolution model simulation that supports latitudinally

alternating zonal jets compared to a lower resolution simulation that does not. Furthermore, Delpech et al. (2020) have noted the importance of these jets for shaping the structure of tracer fields in the tropical Pacific. Brandt et al. (2015) document a multi-decadal downward trend in ETNA oxygen concentration that could be a consequence of anthropogenically driven ocean deoxygenation (Stramma et al., 2008, 2010; Keeling et al., 2010; Schmidtko et al., 2017). Brandt et al. (2015) note, however, that there are also stronger trends, both upward and downward, on the decadal time scale (see Fig. 1b,c taken from Fig. 19 in

Brandt et al. (2015)). It has been suggested that these trends are associated with changes in the zonal jets, and in particular the strength of these jets (Brandt et al., 2010; Hahn et al., 2017).

Latitudinally alternating zonal jets, as seen in Fig. 1a, are an ubiquitous feature of the ocean (Maximenko et al., 2005, 2008). Their identification using Argo float data is shown in Fig. 3 of Ascani et al. (2015) and their central role in the tropical ocean

circulation is illustrated in Fig. 1 of Ménesguen et al. (2019) who also discuss possible mechanisms for supporting these jets.
One of these mechanisms involves the net eddy-induced westward mass flux by mesoscale eddies and argues that the zonal jets are required to provide a net eastward mass flux in order to conserve mass (Marshall et al., 2013).

   Here, we work with the simplest possible dynamical model that can generate mesoscale eddies and that, in turn, supports zonal jets, with the aim to shed light on both the formation of the ETNA OMZ and the decadal to interdecadal variations of oxygen concentration within the OMZ. The model is a non-linear $1\frac{1}{2}$-layer reduced-gravity model and is very similar to that
used by Qiu et al. (2013) to illustrate the generation of zonal jets in the tropical Pacific Ocean through Rossby wave resonant triad interactions. While the model includes the realistic coastlines of the North Atlantic, we note that very similar results are obtained using rectangular basin geometry, as shown by Köhn (2018), and that the effect of varying ocean bathymetry is excluded. The dynamical model is coupled to a simple advection-diffusion model for a passive tracer that includes a tracer source near the western boundary and a simple Newtonian relaxation term to mimic oxygen consumption. By coupling to
the advection-diffusion model, we illustrate how the latitudinally alternating zonal jets can advect oxygen from the western boundary into the interior of the ocean, with their effect diminishing eastwards, facilitating the formation of an OMZ near the eastern boundary. Due to their inherent nonlinearity, originating in the mesoscale eddy field, these jets can exhibit their own internal variability, even on decadal time scales, analogous to what is known about the atmospheric circulation (James and James, 1989). There is, therefore, the potential to influence oxygen concentration in the OMZs on similar time scales.
The structure of the paper is as follows. Section 2 describes the model set-up and forcing, Sect. 3 presents the results and Sect. 4 provides a Summary and Discussion.

## 2   Methods

### 2.1   The model

The governing equations for the dynamic model are those for a non-linear $1\frac{1}{2}$-layer reduced-gravity model, namely:

$$\frac{\partial u}{\partial t} - qhv = -\frac{g'}{a\cos\theta}\frac{\partial E}{\partial \lambda} + M_u \tag{1}$$

$$\frac{\partial v}{\partial t} + qhu = -\frac{g'}{a}\frac{\partial E}{\partial \theta} + M_v \tag{2}$$

$$\frac{\partial h}{\partial t} + \frac{1}{a\cos\theta}\left[\frac{\partial(hu)}{\partial \lambda} + \frac{\partial(hv\cos\theta)}{\partial \theta}\right] = F_h. \tag{3}$$

Here $(\lambda, \theta)$ are longitude and latitude, respectively, $(u, v)$ are the eastward and northward velocities, respectively, $h$ is the active layer thickness, $E = \frac{1}{2}(u^2 + v^2) + g'h$ is the Bernoulli function where $g'$ is the reduced gravity, and $q = (f + \zeta)/h$ is the potential vorticity where $\zeta = \frac{1}{a\cos\theta}(\frac{\partial v}{\partial \lambda} - \frac{\partial u\cos\theta}{\partial \theta})$ is the relative vorticity, $f = 2\Omega sin\theta$ is the Coriolis parameter, $\Omega = 7.29 \times 10^{-5}\mathrm{s}^{-1}$ is

the rotation rate of the earth, and $a$ is the radius of the earth. $(M_u, M_v)$ are the lateral mixing of momentum terms. These are given, following Shchepetkin and O'Brien (1996), by

$$M_u = -\frac{1}{h}\left[\frac{1}{a\cos\theta}\frac{\partial}{\partial\lambda}P_{\lambda\lambda} + \frac{1}{a}\frac{\partial}{\partial\theta}P_{\lambda\theta} - \frac{2\tan\theta}{a}P_{\lambda\theta}\right] \tag{4}$$

and

$$M_v = -\frac{1}{h}\left[\frac{1}{a\cos\theta}\frac{\partial}{\partial\lambda}P_{\lambda\theta} + \frac{1}{a}\frac{\partial}{\partial\theta}P_{\theta\theta} - \frac{2\tan\theta}{a}(P_{\lambda\lambda} - P_{\theta\theta})\right], \tag{5}$$

where

$$P_{\lambda\lambda} = -A_h h\left[\frac{1}{a\cos\theta}\frac{\partial}{\partial\lambda}u - \frac{1}{a}\frac{\partial}{\partial\theta}v - \frac{\tan\theta}{a}v\right], \tag{6}$$

$$P_{\lambda\theta} = P_{\theta\lambda} = -A_h h\left[\frac{1}{a\cos\theta}\frac{\partial}{\partial\lambda}v - \frac{1}{a}\frac{\partial}{\partial\theta}u + \frac{\tan\theta}{a}u\right], \tag{7}$$

and $P_{\theta\theta} = -P_{\lambda\lambda}$, with $A_h$ being the horizontal eddy viscosity. $F_h$ is the model forcing described below.

The model domain spans the latitude/longitude range 20°S to 50°N and 65°W to 15°E, uses the realistic coastlines of the Atlantic Ocean at the western and eastern boundaries, and has a horizontal resolution of 0.1°. The northern and southern boundaries are closed, as is the Strait of Gibraltar, and a no-slip boundary condition is used with a horizontal eddy viscosity of $A_h = 100\text{m}^2\text{s}^{-1}$. The model is set-up with an undisturbed layer depth of $H_0 = 500\text{m}$ and a reduced gravity of $g' = 1.5\times10^{-2}\text{m}$ s$^{-2}$ giving a gravity wave speed $c = \sqrt{g'H_0} = 2.7ms^{-1}$ that is within the range of first baroclinic mode gravity wave speeds for the tropical Atlantic (Chelton et al., 1998). The forcing corresponds to a mass flux in and out of the dynamically active layer that is confined near the equator and oscillates at annual period $\frac{2\pi}{\omega}$, and is given by

$$F_h(\lambda, \theta) = \begin{cases} -A_0\sin(\omega t)\exp[-\beta y^2/(2c)] & \text{if } 15°S \le \theta \le 15°N \\ 0 & \text{otherwise,} \end{cases} \tag{8}$$

where $A_0 = 3.0\text{x}10^{-5}\text{ms}^{-1}$, corresponding to $\approx 2.6\text{m day}^{-1}$, $y$ is the meridional distance from the equator and $\beta$ is the meridional gradient of $f$. The e-folding distance around the equator, $\sqrt{2c/\beta}$, is 4.6° latitude corresponding to the equatorial radius of deformation. This choice of forcing is a simple and efficient way to induce an annual cycle of layer thickness and zonal velocity along the equator (in reality associated with the annual cycle of wind stress) and is not intended to be realistic.

The dynamical model is coupled to an advection-diffusion model for which the governing equation is

$$\frac{\partial Ch}{\partial t} + \nabla\cdot(h\mathbf{u}C) = \nabla\cdot(\kappa_h h\nabla C) - JCh - h\gamma(C - C_0) + CF_h, \tag{9}$$

where $C$ is the tracer concentration (mimicking oxygen), $\kappa_h$ is the tracer diffusivity (set equal, for simplicity, to the eddy viscosity $A_h$) and $\nabla\cdot(...)$ is the divergence operator in spherical (latitude/longitude) coordinates. The tracer source is given by the $h\gamma(C - C_0)$ term where

$$C_0(\lambda, \theta) = \begin{cases} 1 & \text{if } 65°W \le \lambda \le 55°W \\ 0 & \text{otherwise} \end{cases} \tag{10}$$

implying a strong source within $10°$ of the western boundary (because of the use of realistic coastlines, the source is confined to the region north of $10°N$; see Fig. 2d). To prevent a sharp edge in the tracer concentration at the step in the $C_0$ field, the restoring factor $\gamma$ is set up to be spatially dependent and is given by

$$\gamma(\lambda,\theta) = \begin{cases} \gamma_0/2(1 - tanh[\frac{\lambda-\lambda_0}{\lambda_s}]) & \text{if } 65°W \leq \lambda \leq 55°W \\ 0, & \text{otherwise} \end{cases} \tag{11}$$

where $\lambda_0 = 63°$, $\lambda_s = 1°$ and $\gamma_0 = 1/8.35$ day$^{-1}$, indicating a strong restoring within a few degrees of the western boundary. Placing the source near the western boundary is consistent with both the observed oxygen distribution (Fig. 1a) and the notion that the most recently ventilated water is associated with the western boundary current system; see, for example, Fratantoni and Richardson (2006) and Kirchner et al. (2009).

Tracer consumption is modelled using a simple Newtonian relaxation term, the strength of which is determined by the tracer consumption rate $J$. Here $J$ is set to $J = 6.5 \times 10^{-10}s^{-1}$ and is associated with a relaxation time scale of roughly 50 years and is, as a result, much longer than the time scale associated with the source term. This value for the consumption rate is half that diagnosed by fitting a one-dimensional advection-diffusion model to observations in the Pacific by Van Geen et al. (2006) and used by Brandt et al. (2010, 2012) in studies of the oxygen field in the equatorial Atlantic over the depth range $300 - 500m$. Using a smaller value ensures that the tracer is able to cross the entire model basin without being damped out and is also consistent with the notion of a lower apparent oxygen utilisation rate at greater depth (Karstensen et al., 2008) where the model is likely to be most applicable. It should be noted that the use of a Newtonian relaxation term, with a constant, spatially uniform consumption rate, is a simplification, sufficient for our purposes in the simple model set-up used here, but, nevertheless a crude representation of the complex biogeochemical processes that determine oxygen consumption in reality. It follows that care should be taken when interpreting the tracer modelled here as a proxy for oxygen.

The equations are discretised on an Arakawa C-grid (Arakawa, 1972) using an Adams-Bashforth $3^{rd}$ order time stepping scheme. The details can be found in Köhn (2018). The model is initialised with a state of rest with the undisturbed layer depth of $H_0 = 500m$ and zero tracer concentration $C$ and is then run for $400$ years with a time step of $500s$.

## 2.2 Thickness-weighted averaging

The most natural way to average the tracer Eq.(9) is to use thickness-weighted averaging (e.g. De Szoeke and Bennett (1993), Greatbatch and McDougall (2003)) where, for any variable $\alpha$,

$$\hat{\alpha} = \frac{\overline{\alpha h}}{\overline{h}} \tag{12}$$

and

$$\alpha = \hat{\alpha} + \alpha''. \tag{13}$$

Here an overbar refers to a long time average and we note that $\overline{h\alpha''} = 0$. Averaging Eq.(9) then gives

$$\frac{\partial \hat{C}\overline{h}}{\partial t} = -\nabla \cdot (\overline{h\mathbf{u}C}) + \nabla \cdot (\overline{\kappa_h h\nabla C}) - J\hat{C}\overline{h} - \gamma\overline{h}(\hat{C} - C_0) + \overline{CF_h}. \tag{14}$$

The first term on the right hand side of Eq.(14) can be understood by noting that the thickness-weighted mean velocity, $\hat{\mathbf{u}}$, is the primary velocity variable after averaging (Greatbatch and McDougall, 2003) and can be decomposed into two parts

$$\hat{\mathbf{u}} = \overline{\mathbf{u}} + \mathbf{u}^* \tag{15}$$

where $\overline{\mathbf{u}}$ is the Eulerian mean velocity and

$$\mathbf{u}^* = \frac{\overline{\mathbf{u}'h'}}{\overline{h}} \tag{16}$$

is the eddy-induced velocity, and here $h = \overline{h} + h'$. The eddy-induced velocity is analogous to the Stokes drift associated with surface gravity waves (Stokes, 1847) and is an additional velocity that needs to be added to the Eulerian mean velocity to account, after averaging, for the Lagrangian advection of tracer (see Marshall et al. (2013) for further discussion). The first term on the right hand side of Eq.(14) can now be decomposed as

$$\nabla \cdot (\overline{h\mathbf{u}C}) = \nabla \cdot (\overline{h}\hat{\mathbf{u}}\hat{C}) + \nabla \cdot (\overline{h\mathbf{u}''C''}) = \nabla \cdot (\overline{h}\,\overline{\mathbf{u}}\hat{C}) + \nabla \cdot (\overline{h}\mathbf{u}^*\hat{C}) + \nabla \cdot (\overline{h\mathbf{u}''C''}). \tag{17}$$

The three terms on the far right hand side correspond to the divergence of the advective flux associated with the Eulerian mean velocity, the eddy-induced velocity, and the transient eddies (the eddy-mixing term), respectively.

## 3 Results

In the model, the annual period forcing along the equator generates annual period equatorial Rossby and Kelvin waves that, in turn, excite annual period coastal Kelvin waves that propagate poleward along the eastern boundary of the model domain. These, in turn, radiate annual period Rossby waves into the interior. These Rossby waves destabilise by means of a resonant triad instability (see Qiu et al. (2013) where a detailed discussion can be found) and shed eddies that subsequently propagate westward towards the western boundary. Figure 2 shows snapshots at 6, 60 and 600 months into the model simulation. The top panel shows the layer depth, $h$, corresponding dynamically to the pressure, and the lower panels show the tracer concentration, $C$. After 6 months, the front excited by switching on the forcing is clearly visible in Fig. 2a extending southwestwards towards the equator from the eastern boundary. The much greater westward extension towards the equator is because the westward propagation speed for (long) Rossby waves is given by $\beta g' h / f^2$ (Anderson and Killworth, 1979) and hence increases rapidly towards the equator where $f = 0$. At later times (Fig. 2b,c), this front has destabilised and is shedding eddies that propagate westward and already, at 600 months fill the model domain. Concurrent with these developments (lower panels), the tracer is confined entirely to the source region near the western boundary at 6 months (Fig. 2d). At 60 months (Fig. 2e), the eddies are already penetrating the western boundary region leading to eastward spreading filaments of relatively high tracer concentration into the interior, away from the western boundary. At 600 months (Fig. 2f), these filaments fill the basin to the west of the destabilised front and are clearly spreading the tracer into the interior of the model domain away from the western boundary. An interesting point is that while the eddies propagate westward, the tracer filaments extend eastward. As discussed by Marshall et al. (2013), the westward propagation of the eddies is associated with a net westward mass flux associated with the eddy-induced velocity (see Eq. (16) and the discussion thereon). Marshall et al. (2013) note that in order to conserve mass, there must

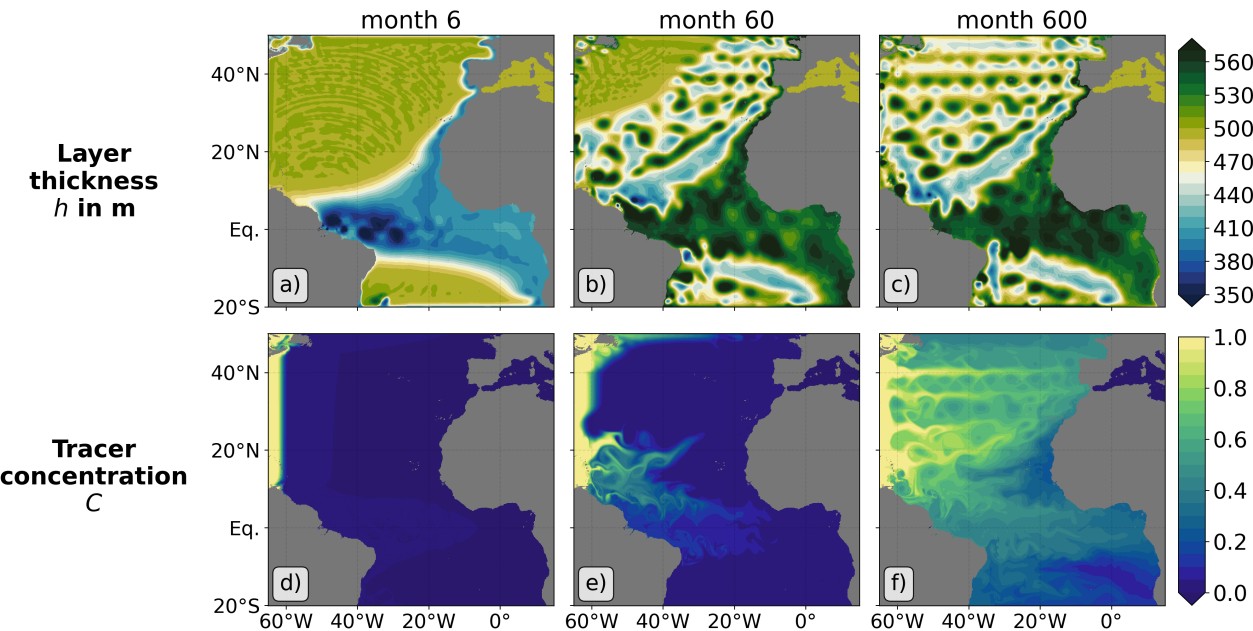

**Figure 2.** Spin-up showing (a-c) the layer thickness, $h$, and (d-f) the tracer concentration, $C$, after 6, 60, and 600 months, respectively.

.

be a corresponding eastward mass flux, and it is this eastward mass flux that manifests itself through the eastward extending filaments. Another interesting point is the advection of tracer along the Rossby wave front at the location where this front starts breaking down into eddies. This can be clearly seen in Figure 2f where the eastward spreading filaments turn northeastward near 20°N.

Figure 3 shows the mean fields averaged over the last 80 years of model integration, that is years 321 to 400. The tracer distribution shown in Fig. 3b is not dissimilar to that shown in Fig. 1a and, intriguingly indicates the presence of a tracer (oxygen) minimum zone east of the destabilising front noted in Figs. 2b,c. This region of tracer minimum is remarkably similar in appearance to the observed OMZ shown in the Fig. 1a being almost exactly colocated with the observed OMZ. In particular, the concentration reaches its minimum southeast of the Cape Verde Islands, hugs the African coast and increases again towards the equator. In the model, its presence is due to the lack of penetration into this region of tracer filaments spreading from the west. The mean thickness-weighted zonal velocity $\hat{\mathbf{u}}$ shown in Fig. 3c shows alternating bands of eastward and westward flow, corresponding to the latitudinally alternating zonal jets noted in the introduction. The corresponding zonal bands in the mean layer thickness reflect the meridionally varying pressure field required to geostrophically balance the zonal jets. These jets are strongest west of the destabilising Rossby front, as we expect, and less coherent at the latitude band of the tracer minimum on the eastern boundary. The mean meridional velocity shown in Fig. 3d is very different and is of much smaller magnitude over most the model domain, and is important only very close to the western boundary and in the western part of the equatorial

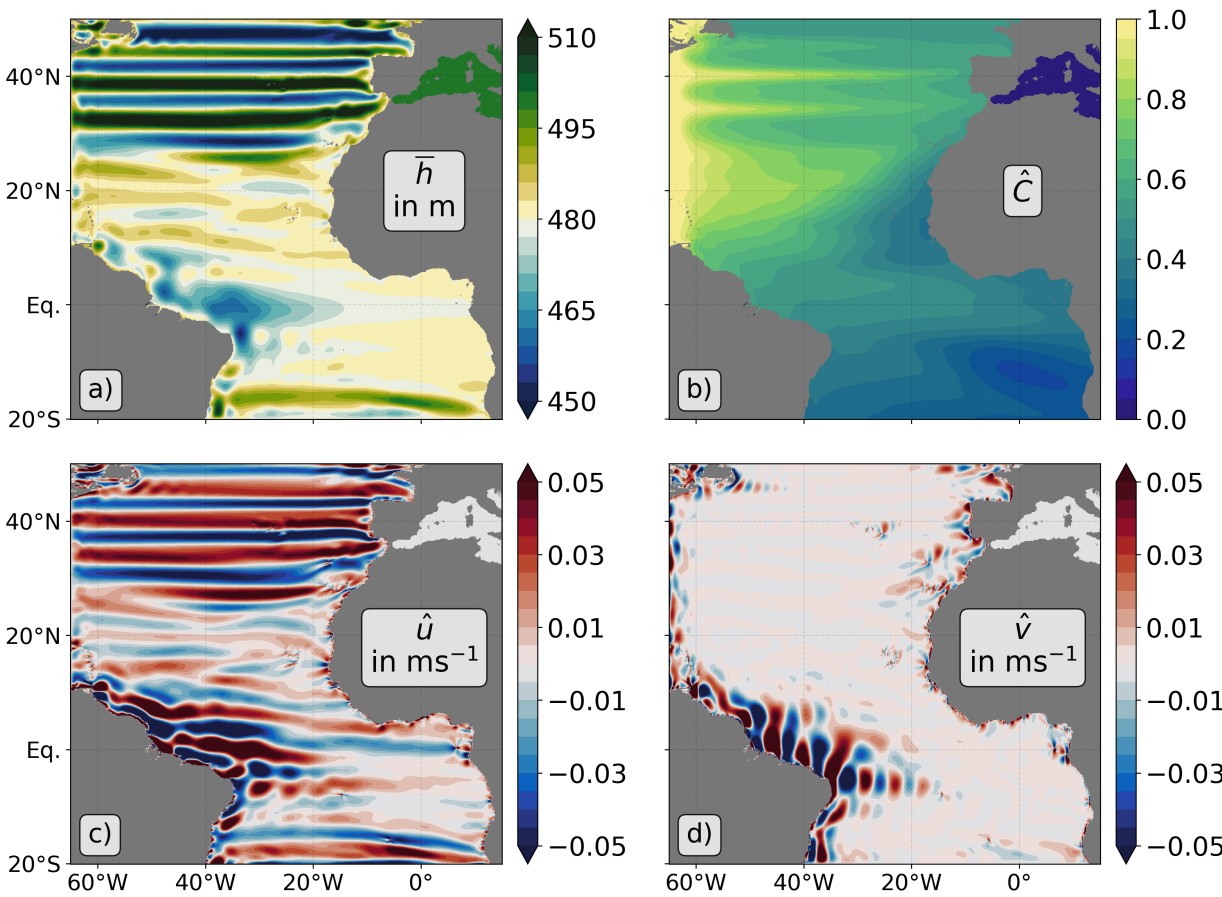

**Figure 3.** Mean fields of (a) layer thickness, $h$, (b) the tracer concentration and (c) the zonal and (d) meridional velocity components over the last 80 years of model integration. The layer depth is averaged in a normal sense, while (b), (c) and (d) show thickness-weighted averages.

.

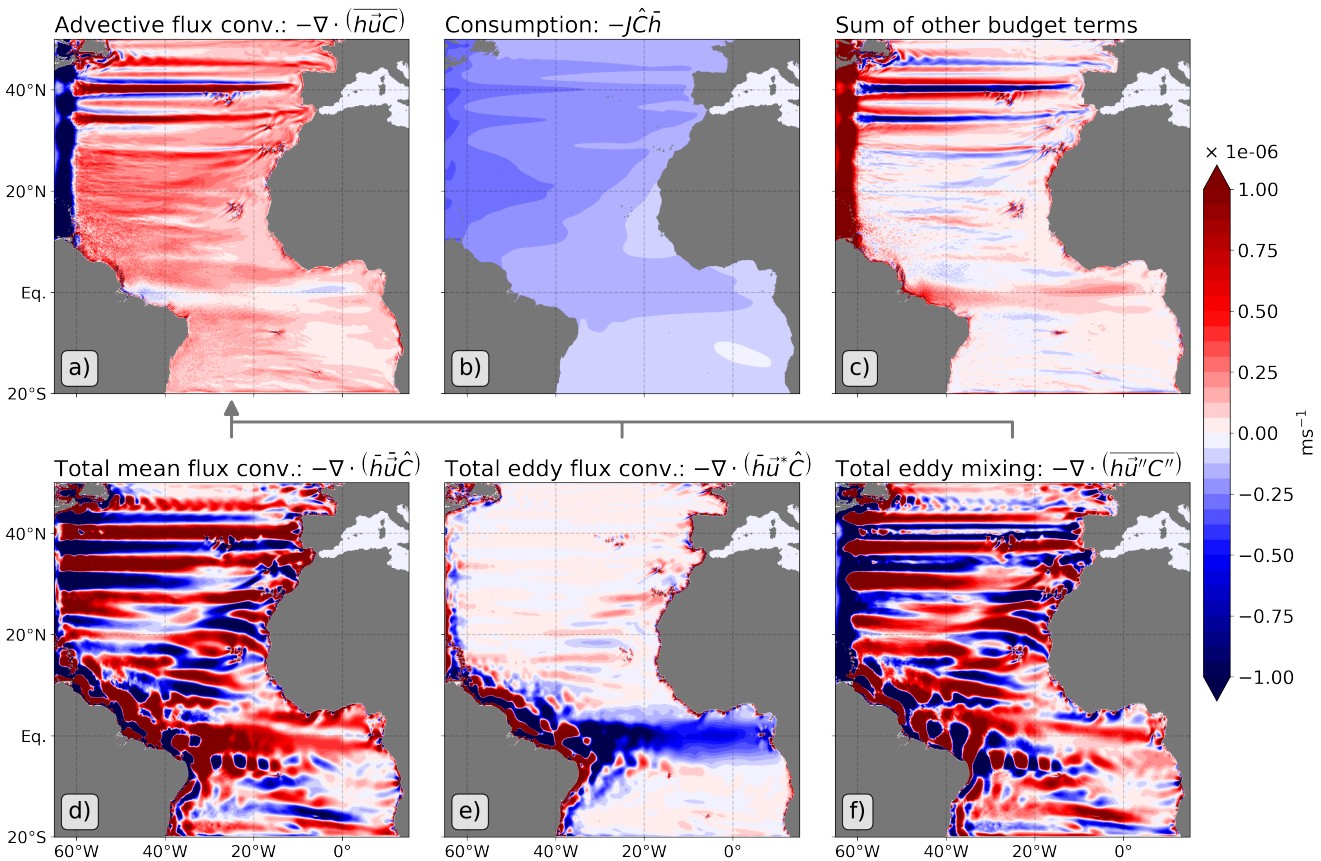

**Figure 4.** Tracer budget terms for the last $80$ years of model integration. See the panel labelling for the individual panel content. The decomposed advective terms in (d), (e) and (f) are summed in (a) following Eq.(17).

.

region, most likely associated with the rectification of (short) Rossby waves that are excited on the western boundary and have eastward group velocity.

A detailed analysis of the tracer budget for the same $80$ years shows that the main balance is between the divergence of the total advective flux, given by the first term on the right hand side of Eq.(14), and the consumption term (see Fig. 4a,b), the local time derivatives playing a negligible role, consistent with the model being in a statistically steady state. This balance is particularly well-established in the latitude band between the equator and $30°$N where the tracer minimum is located. Splitting the divergence of the total advective flux into its separate parts shows that the dominant terms are associated with the Eulerian mean velocity, $\nabla \cdot (\bar{h}\overline{\mathbf{u}}\hat{C})$ (Fig. 4d), and the transient eddies, $\nabla \cdot (\overline{h\mathbf{u}''C''})$ (Fig. 4f), with these two terms showing a strong tendency to cancel each other. The term associated with the eddy-induced velocity (Fig. 4e) is of much smaller magnitude. Nevertheless, as can be seen from Fig. 4a, the net effect of the advective tracer flux is to provide a source of tracer. The

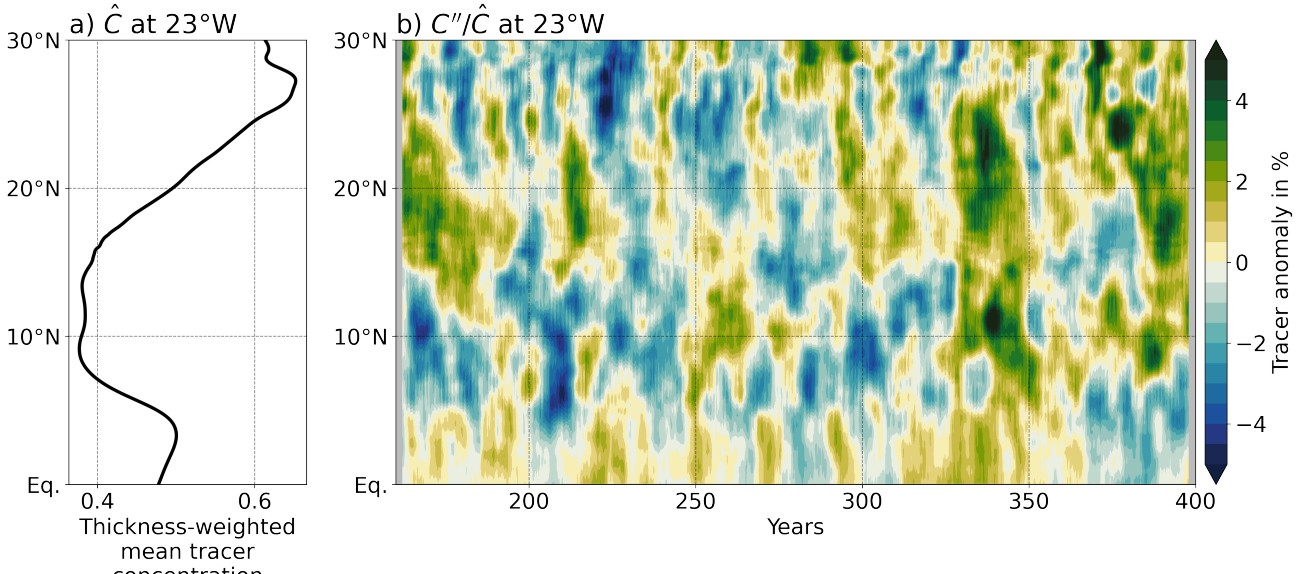

**Figure 5.** (a) Thickness-weighted average of tracer concentration along $23°$W over the last 240 years of model integration and (b) the temporal evolution of the corresponding anomalies in tracer concentration, shown as a percentage of the mean concentration shown in (a). The anomalies are convolved with a 4 year (i.e. 49 month) kernel to remove the seasonal cycle.

.

importance of the Eulerian mean velocity and the transients is, perhaps, not surprising when looking at the instantaneous snapshot in Fig. 2f, with filaments extending eastwards from the western boundary region but also being advected meridionally by the transient eddies.

Although the dynamical model is forced only at the annual period, being a nonlinear model, it nevertheless exhibits variability on much longer (decadal to interdecadal) time scales (James and James, 1989). This low frequency variability is reflected in the tracer concentration, as can be seen in Fig. 5b. Variations in tracer concentration of up to $5\%$ from the mean take place on decadal to interdecadal time scales. Köhn (2018) carried out a detailed analysis of the trends in the model and concluded that the observed trends in the ETNA OMZ, namely those discussed by Brandt et al. (2015) and Hahn et al. (2017), are largely within the range of trends in the model. The two major decrepancies are the pronounced oxygen decrease in the upper (150-300m depth) OMZ during 2006-2013 (an $\approx 14\%$ decrease) and the multidecadal decrease in the lower (350-700m depth) OMZ (an $\approx 14\%$ decrease, this time over 50 years) that can be seen in Fig. 1b and 1c, respectively. It should, nevertheless, be remembered that the model used here is very idealised and that results from much more complex models may lead to different conclusions. The important point is the presence of decadal to multidecadal trends in the tracer concentration exhibited by the statistically steady state of the model. These trends are driven by changes in the advective forcing of the tracer concentration, consistent with the suggestion of Brandt et al. (2010).

## 4 Summary and discussion

Calil (2023) has shown that using a high resolution ocean model to resolve the latitudinally alternating zonal jets leads to a much better representation of the Eastern Tropical North Atlantic (ETNA) Oxygen Minimum Zone (OMZ) than when these jets are not resolved. Earlier, Brandt et al. (2010) had suggested that changes in these jets, in particular their strength, might be able to explain decadal trends in the oxygen concentration within the ETNA OMZ. Here, we have presented results from a simple dynamical model that supports mesoscale eddies and the corresponding latitudinally alternating zonal jets with a view to shedding light on both the formation of the ETNA OMZ and the subsequent decadal to interdecadal variability of oxygen concentration within the OMZ. The simple dynamical model, which uses the geometry of the North Atlantic Ocean, is coupled to an advection-diffusion model to mimic oxygen. The tracer carried by the advection-diffusion model has a source near the western boundary of the North Atlantic and a relaxation term to mimic consumption. The results show that filaments of relatively high tracer concentration are carried by the Eulerian mean jets and the eddies eastward away from the western boundary source towards the eastern boundary. The mean tracer distribution in statistically steady state is remarkably similar to that in the observed ocean with a minimum in tracer concentration in the same general location as the observed OMZ. This region of tracer minimum is situated to the east of the Rossby wave front which destabilises and is the source of the eddies in the model, exactly as described by Qiu et al. (2013).

We noted in the introduction that the OMZs in the ocean have been associated with the shadow zones that are predicted by the ventilated thermocline theory of Luyten et al. (1983) for the wind-driven circulation of the subtropical gyre, regions that in the theory are not ventilated and are thought to be characterised by a near-stagnant ocean. We note that in the theory, the ventilated thermocline is driven by the downward Ekman pumping due to the wind stress curl over the subtropical gyre. Nevertheless, the simple model we have described here excludes this downward Ekman pumping and is forced only near the equator, with no forcing in the latitude band of the tracer minimum shown in Fig. 3b, yet exhibits an OMZ remarkably similar to the observed OMZ in the tropical North Atlantic. Indeed, the mechanism we have demonstrated offers an alternative explanation for the existence of OMZ's, independent of the ventilated thermocline theory. It is an interesting point that the ETNA OMZ we have focused on is located in a region where the mean wind stress curl has the opposite sign to that associated with the subtropical gyre (see, for example, Leetmaa and Bunker (1978) or the more recent estimate by Risien and Chelton (2008)). It follows that the ventilated thermocline theory, which relies on the equatorward Sverdrup transport associated with a subtropical gyre, is unlikely to be applicable to the ETNA OMZ. However, this is not to say that wind forcing has no role to play in explaining the observed ETNA OMZ. In particular, Peña-Izquierdo et al. (2015), while noting the importance of zonal jets for ventilating the ETNA OMZ, note also that the upper OMZ is ventilated predominantly by water from the South Atlantic via the North Equatorial Countercurrent (NECC), a wind-driven current, and the North Equatorial Undercurrent (NEUC). Likewise, the lower OMZ was found to be ventilated prodominently by water originating in the North Atlantic subtropical gyre by means of deep zonal jets (Brandt et al., 2015) analogous to those simulated by our model. It follows that both the mechanism we have identified here, and the wind-driven circulation, play a role in ventilating the ETNA OMZ in reality, but that the shadow zone predicted by the ventilated thermocline theory is not required to explain the ETNA OMZ. It also seems likely that the

latitudinally alternating zonal jets associated with the mesoscale eddy field, such as we have discussed here, are most applicable for ventilating the lower OMZ, below roughly 250m depth, with the direct effect of the wind-driven circulation being more important for ventilating the upper, roughly 250m, of the water column. We also note that although the ventilated thermocline theory of Luyten et al. (1983) may not be relevant for explaining the ETNA OMZ, this does not mean that it does not play a role in the dynamics of other OMZs, in particular those in the Pacific Ocean.

We have shown that although the dynamical model has forcing only on the annual time scale, the modelled tracer concentration exhibits pronounced decadal and multidecadal variability. The analysis by Köhn (2018) suggests that the associated decadal and interdecadal trends offer a plausible explanation for the observed trends in the ETNA OMZ, for example those documented by Hahn et al. (2017). Exceptions are the slow, multidecadal decrease in oxygen in the lower OMZ (350-700m depth) and the rapid decrease in the upper OMZ (150-300m depth) observed between 2006 and 2013 (see Fig. 1b,c taken from Brandt et al. (2015)). However, it should be remembered that the model we have used is very simplified and that further studies using more comprehensive models are needed. Nevertheless, we note that more comprehensive models generally underestimate rates of multidecadal oxygen decline, as well as interannual to decadal variability (Oschlies et al., 2018). One possible reason for this, supported by the work of Calil (2023), may well be an inability of the models to properly resolve the latitudinally alternating zonal jets.

Finally, we note that Chang et al. (2008) have argued that the amplitude of the annual cycle in the equatorial Atlantic can be expected to reduce as the strength of the Atlantic Meridional Overturning Circulation reduces under anthropogenic forcing. While acknowledging that the forcing we use to drive our model is very idealised, this suggests that a future climate scenario might be mimicked by reducing the amplitude of the forcing applied to the model. Qiu et al. (2013) have shown that in this case, the front, where the westward propagating Rossby waves destabilise, moves westward. Such a westward displacement would, in turn, reduce the ventilation of the OMZ in our model, expanding the OMZ westward. It is possible, therefore, that the ventilation of the OMZs in the ocean by the latitudinally alternating jets will be reduced in the future, contributing to the deoxygenation of the ocean that is expected as a result of anthropogenic climate change (Stramma et al., 2008, 2010; Keeling et al., 2010; Schmidtko et al., 2017).

*Code and data availability.* Scripts for data processing and plotting can be found at https://doi.org/10.5281/zenodo.11447291. CMEMS GLORYS12v1 data can be retrieved from https://doi.org/10.48670/moi-00021. World Ocean Atlas 2018 oxygen data can be retrieved from: https://www.ncei.noaa.gov/archive/accession/NCEI-WOA18. Model output and code (the tarball swm.tar.gz) are stored and accessible at
https://hdl.handle.net/20.500.12085/dd331654-413c-4157-8796-6edf4c4be207

*Author contributions.* All authors contributed to the conceptualisation and methodology. MC wrote the model code. EK carried out the formal analysis with help from MC. RJG prepared the manuscript with contributions from all co-authors.

*Competing interests.* The authors declare that they have no conflict of interest.

*Acknowledgements.* Funding from the Deutsche Forschungsgemeinschaft as part of Sonderforschungsbereich 754 "Climate-Biogeochemistry Interactions in the Tropical Ocean" is gratefully acknowledged. This study has been conducted using E.U. Copernicus Marine Service Information; (https://doi.org/10.48670/moi-00021). We are grateful to Klaus Getzlaff for his help with the data archive generated by the model and to two anonymous reviewers for their helpful comments on the manuscript.

265

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
