# Peer review of "The formation and ventilation of an Oxygen Minimum Zone in a simple model for latitudinally alternating zonal jets"

_EGUsphere, 2024_

## Author Comment (AC1)

Thank you for your very positive and encouraging review. Below we list your suggestions for improvements and our response.

1. While the model is quite suited for the question, I am not sure how directly interpretable the tracer is as oxygen. The model does not include biogeochemical processes such as remineralisation, which are also important for the formation of the oxygen minimum zones. While I do not think the authors should try extending their model with biogeochemistry, I do think that they should be more explicit (especially in the abstract and introduction) that they only consider the advective effect of an idealised tracer, and that care should be taken when interpreting the tracer as a proxy for oxygen.

In the abstract, introduction and the methods section, we have added some material to clarify that we work with a passive tracer for which the consumption is represented by a Newtonian damping term. We note in the methods section that this is crude representation of the biogeochemical processes that lead to oxygen consumption but that it is sufficient for our purpose here in a simple model setting.

2. I don't fully understand why the model is 'only' 0.1 degrees resolution. This is barely eddy-resolving, and indeed the eddies in Figure 2 look conspicuously circular and uniform. For what seems to be a quite efficient model (no vertical dimension), why not increase the resolution and also resolve more of the submesoscale processes?

Since our model only has one active layer, the range of possible instability mechanisms is actually quite limited. The instability mechanism in our model set-up is the same as that described by Qui et al. (2013) and is a resonant triad mechanism for Rossby waves that is well-resolved by our model. However, a single layer model such as ours does not support submesoscale processes and, indeed, increasing the resolution is unlikely to make any significant difference to the model results.

3. One of the most interesting results to me is that the tracer spreading is eastward; against the (I assume) westward propagation of the eddies. Can the authors more carefully discuss _how_ the eddies would spread tracer in the opposite direction as their translation.

We have added some material on this to the results section where we briefly discuss this issue. As noted by Marshall et al. (2013), the westward propagating eddies are associated with a net westward mass flux that must be balanced by a corresponding eastward mass flux in order to conserve mass. The eastward mass flux manifests itself as the eastward spreading filaments in our model results.

4. The authors nicely contrast the role of eddies with the ventilated thermocline theory, but then stop short of arguing which is more important in the real ocean. Can they not explore the relevance of both methods a bit more? That would greatly help the community

In the revised manuscript, we have expanded the discussion on this topic in the Summary and discussion section. We note that the Eastern Tropical North Atlantic Oxygen Minimum Zone (ETNA OMZ) actually sits to the south of the subtropical gyre in the North Atlantic and so is unlikely to be simply associated with a shadow zone as predicted by ventilated thermocline theory. Nevertheless, as we note in the text, there is evidence that the ventilation of the ETNA OMZ involves both zonal jets and aspects of the wind-driven circulation. The paper by Pena-Izquierdo et al. (2015) is particularly relevant to this discussion.

I also have some minor comments:

- lines 2 and 14: The abstract and introduction both start a bit sudden with the method; it is typical to first motivate the study by introducing its relevance?

A couple of sentences have been added to the start of the main text by way of introduction. We have not done this in the abstract where we need to be careful not to exceed the 250 word limit.

- Figure 1 (panel an and caption): Why is this field given at ~500? Why not provide the exact depth?

What is shown is an average of data from 454m and 541m depth as we now note in the figure caption.

- End of introduction: It is interesting that the bathymetry apparently does _not_ play a role in the organisation of the zonal jets. I had seen other studies in the past (I think?) that argued that the location of zonal jets was 'pinned' by bathymetry? Perhaps the authors want to reflect why their model without bathymetric effects (except coastlines) still has zonal jets?

We now note towards the end of the introduction that our model excludes the effect of varying ocean bathymetry. As note in the text, the instability mechanism in our model is the same as that described by Qui et al. (2013) and is a resonant triad mechanism that functions independent of the ocean bathymetry.

- line 73: be explicit in which direction the mass flux is?

The mass flux alternates at annual period and is directed into and out of the active upper layer of the 1 ½ layer model we use. The text has been adjusted so that we state this explicitly in the revised version.

- line 80: give a motivation/reference why the tracer diffusivity can be set equal to the eddy viscosity?

This is done only for simplicity, as we now note in the text.

- line 90: why is the consumption rate only half? Is there an intuitive reason?

As we note in the text, the value for the consumption rate we use ensures that the tracer spreads across the whole basin in the model run and is consistent with the notion of a lower apparent consumption rate at depth, as noted by Karstensen et al. (2008).

- line 112: give an intuitive explanation why the eddy-induced velocity here is analogous to Stokes drift?

We have added some material to hopefully clarify this point. In particular, the eddy-induced velocity is an additional velocity that needs to be added to the Eulerian mean velocity to account, after averaging, for the Lagrangian advection of tracer. We refer to Marshall et al. (2013) for further discussion.

- line 116: maybe I missed it, but what are the boundary conditions on the open boundaries for the dynamic model?

In the methods section, it is stated that these boundaries are closed.

- line 135: Can 'remarkably similar' here be quantified?

We have not attempted to quantify what we mean by "remarkably similar" but we have added some material to back-up our assertion.

- line 143: This role of rectification of short Rossby waves could be tested/assessed? It doesn't need to be assumed?

It is not easy to say exactly what the features are in the mean meridional velocity field shown in Figure 3d. However, it is clear that the oscillating forcing around the equator will generate short Rossby waves at the western boundary. What one sees in the figure look like standing short Rossby waves that result from this process. We have not tried to dig any deeper into what is exactly going on here since it is not central to the topic of the manuscript.

- line 216: I see that the data of the model is available via a dot, but could not easily

locate the modelcode itself. In the spirit of open and reproducible science, I wonder whether the code of the model is available somewhere?

In the "Code and data availability" section it is stated "Model output and code are stored and accessible at….". We have now added that the code is in the tarball swm.tar.gz

---

## Author Comment (AC2)

Thank you for your very positive and encouraging review. We respond to your suggested revisions below.

1.  Explain better the physical interpretation of the annual mass flux driving the physical model. Does this implicitly represent the seasonal cycle of wind forcing in some way? Be more explicit about how the forcing differs in the conception of the OMZ as a consequence of ventilated thermocline theory. How might the results differ with climatological wind stress forcing instead?

In the methods section we note that the forcing we use is an oscillating mass flux into and out of the active layer of our model. It is a simple way to induce an annual cycle in the equatorial region that, in reality, is associated with the annual cycle of wind stress, but, as we note in the revised text, is not intended to be realistic. Later, in the summary and discussion section, we have expanded the discussion comparing our results to the ventilated thermocline theory approach. We note that the forcing we use to drive our model excludes wind forcing for a subtropical gyre, as required by that theory, and is confined near the equator with no forcing in the latitude band of the tracer minimum shown in Figure 3b. We also note that the observed East Tropical North Atlantic Oxygen Minimum Zone sits south of the subtropical gyre in the North Atlantic and so is unlikely to be associated with a shadow zone as in the ventilated thermocline theory. To be honest, we do not see what would be gained by running our model with climatological wind forcing. For one thing, the ventilated thermocline theory requires a minimum of two active layers and so is not supported by the simple 1 ½ layer model we use.

2.  Explain the rationale behind the oxygen source term better. Perhaps make a comparison between Fig3b and Fig1a more explicitly.

We have added some material on this when the forcing is introduced in the methods section, noting the motivation from Figure 1a and also the notion that the most recently ventilated water is associated with the western boundary current system. Also, when discussing Figure 3 we now note the similarity between the tracer distribution shown in Figure 3b and the oxygen field shown in Figure 1a.

3.  Consider revising the beginning of the introduction. I appreciate that the paper is concise, but starting the paper with "Figure 1a…" is a bit over the top.

A couple of sentences have been added by way of introduction.